# Experimental Study on Wave Characteristics of Stilling Basin with a Negative Step

**DOI:** 10.3390/e24040445

**Published:** 2022-03-23

**Authors:** Guibing Huang, Mingjun Diao, Lei Jiang, Chuan’ai Wang, Wang Jia

**Affiliations:** State Key Laboratory of Hydraulics and Mountain River Development and Protection, Sichuan University, Chengdu 610065, China; 2019223069156@stu.scu.edu.cn (G.H.); jianglei@scu.edu.cn (L.J.); 2020323060033@stu.scu.edu.cn (C.W.); jiawang@stu.scu.edu.cn (W.J.)

**Keywords:** stilling basin, negative step, hydraulic model, wave characteristics

## Abstract

Stilling basin with a negative step is an important structure in hydraulic systems, because it can avoid atomization and decrease scouring problems. Although stilling basins with a negative step have attracted much attention from researchers, few researchers have focused on the wave characteristics. In this research, an experimental study on the wave characteristics of stilling basins with a negative step was carried out. The wave height, average period, wave probability density and power spectrum along the flow direction of different stilling basins with a negative step were described based on the wave theory, and the results indicate discharge and step height have a significant effect on the wave characteristics. The relationships between the different characteristic wave heights, and the empirical formula for the relative characteristic wave height are obtained. Finally, the dimensionless standard deviation at the end of the stilling basin with a negative step is linearly related to the flow-energy ratio and the relative step height under B-jump.

## 1. Introduction

With the increasing awareness of environmental protection, the construction of high dam projects has to consider the eco-friendly regulation. Based on this background, the stilling basin with a negative step is proposed. For a high dam project, the stilling basin with a negative step is used for energy dissipation to effectively solve the problems of the fluctuating pressure of the floor, the excessive flow velocity near the bottom of the stilling basin [1], and atomization so that the impact on the environment is smaller than the traditional energy dissipation by hydraulic jump and ski-jump energy dissipation. Therefore, it has been widely used in many hydropower projects in the world, Table 1 gives the representative large hydropower project in the world using the stilling basin with a negative step.

The so-called stilling basin with a negative step is a stilling basin with a step set at the entrance that makes the end of the spillway higher than the bottom surface of the stilling basin. Although great progress has been made in theoretical analysis and experimental research on the hydraulic characteristics of the stilling basins with a negative step, most of the research focuses on flow pattern [2,3,4], the energy dissipation mechanism [1], flow velocity near the bottom [5,6], floor stability [7], fluctuating pressure [8,9] and geometric parameter optimization [10]. However, when large-flow and high-dam projects use stilling basins with a negative step, violent waves are prone to appear. If not taken seriously, this may result in incalculable damage to the entire project. In Wu’s [11] experiment, it was discovered that the post jump waves caused by energy dissipation by hydraulic jumps can reach several meters and spread far downstream. Since the completion of the Xijin Hydropower Station [12], both sides of the downstream banks have often been threatened by waves.Based on the results of the Xiangjiaba stilling basin hydraulic model test, Li [13] once pointed out the stilling basin with a high negative step is prone to large-scale water surface fluctuations. In actual engineering, if this type of problem is not properly resolved, large waves may severely scour the downstream bank slopes. For instance the long-distance propagation of waves can affect downstream navigation and power generation by power plants, and wave reflection and climbing can directly affect the normal operation of the project.

The main function of the stilling pool is to dissipate water energy and to control water jumps in the stilling basin. The water flow in the hydraulic jump is violently turbulent, and a large-scale vortex structure is generated. The more uneven the flow velocity distribution in the stilling basin is, the greater the wave generated. Wu et al. [11] studied the height, period and spectrum of waves after hydraulic jumps and deduced the relationship between the maximum wave height (*H*_max_) and the length of the hydraulic jump. In addition, Dong et al. [12] studied the water surface fluctuation characteristics of energy dissipation of surface regime, and their test results showed that the flow pattern in the stilling basin is closely related to the magnitude of wave fluctuation. Zheng et al. [14] studied the variation law of water level in the stilling basin under different step heights and flow pattern transformation. Waves are a form of movement of water, as well as a form of the expression of energy. In the past, stilling basins were mainly used for low-head flood discharge and energy dissipation, and the wave fluctuations in the stilling basin were small and did not affect the safe operation of the project; therefore, there are few studies on wave research in stilling basins. However, a stilling basin with a negative step is mainly used in high-head and large-flow flood discharge projects, the water surface fluctuates drastically during flood discharge, and the wave problem cannot be ignored. Although there are a few studies on the waves in stilling basins and there is not enough information available for reference, at present, wave-related theories have become more mature and have played an important guiding role in ocean engineering and other fields.

According to the wave theory, waves are considered to be highly irregular and random processes in space and time [15]. In engineering fields, people pay more attention to the basic elements of waves, including the height, period, and wavelength. At present, the zero-crossing method is commonly used to define the height and period of a wave. The wave height is the most important of the wave elements. Many scholars have studied the time-domain distribution [16,17], probability distribution [18], and prediction [19,20] of characteristic wave heights, the wave-height statistics [21]. Some scholars have also proposed theoretical expressions for wave height distribution. For example, Longuet-Higgins [22] not only found that the distribution of wave heights under steady sea conditions is near the distribution of a stationary random Gaussian process but also rigorously derived the probability distribution of the maximum wave height (*H*_max_). Tayfun [23] proposed a theoretical expression for the statistical distribution of wave heights and acquired asymptotic approximations of the probability density of large wave heights on the basis of previous theories by Longuet-Higgins [24], Naess [25], and Vinje [26]. The wave period is also the focus of wave research. Paolo Boccotti [17] proved that quasi-determinism (QD) theory is valid in evaluating the probability of the period of waves in shallow and deep waters. Most scholars combine the wave period with the wave height, amplitude and other elements, such as Tayfun [27], who deduced the theoretical formula for the simultaneous distribution of the large wave heights and periods, and the associated probability distribution for wave heights and periods are optimized by Pual Stansell [28], who also derived an expression for the joint distribution of the wave amplitude and period. Huang [29] studied the characteristics of roll-wave based on numerical investigation, such as wave–wave interactions, generality, and spectrum. In terms of wave propagation, Ling studies [30] the performances of theoretical wave attenuation models in predicting wave damping caused by vegetation. Wei [31] investigated the influence of uniform vegetation on wave attenuation through physical model experiments, while Blackmar [32] investigated wave high attenuation in non-uniform vegetation by physical model experiments and numerical simulation. The research methods on waves are very mature, but most of the research on waves has focused on ocean waves. Few scholars have studied stilling basins on the basis of on wave theory, in particular, the waves in stilling basins with a negative step.

In the past, the measurement of waves or water surfaces in stilling basins was mostly performed with point gauges or pressure measuring tubes. Due to the severe fluctuation of the water surface, the measurement results were inaccurate and random. In this study, a digital wave elevation gauge with high accuracy and frequency was used to measure waves. The measurement time met the ergodic theory of various states. The analysis method was based on wave theory, which guaranteed the reliability of the test data and test results. This paper took a hydraulic project as an example to carry out hydraulic model tests of a stilling basin with a negative step. The aim of this paper is to (1) analyze the wave height, average period, wave probability density and power spectrum; (2) obtain the relationship of different characteristics wave height and the wave height forecasting methods; (3) discuss the influence of the step heights and flow-energy ratio on the wave characteristics at the end of a stilling basin with a negative step.

## 2. Experimental Setup and Data Processing

### 2.1. Facilities of the Model Tests

The experimental facilities included a flow measurement weir, a reservoir, a spillway (0.5 m wide), a stilling basin with a negative step (0.5 m wide) and a circulating water supply. To facilitate the observation of the flow state of the water, the stilling basin with a negative step and spillway were made of transparent plexiglass. The geometric parameters of the stilling basin with a negative step in this test were designed according to those of a water conservancy project, and the hydraulic model test was carried out at a ratio of 1:80. The step heights of the stilling basin with a negative step are shown in Table 2. A total of 3 stilling basins with a negative step (TP1~TP3) and a standard stilling basin (TP0) were included. The height of the dam, the length of the stilling basin and the height of the end sill were all the same for each test (dam height *P* = 1.536 m, stilling basin length *L* = 2.125 m, and end sill height *c* = 0.2625 m). The flow discharges *Q* of each test were 30 L/s, 50 L/s, 70 L/s and 90 L/s, and a total of 16 tests were carried out. The experimental setup is shown in Figure 1.

The wave signal was measured by means of a digital wave elevation gauge (SDA1000, Chengdu Yufan Technology, Inc., Chengdu, China). The sensor model was YWH200-D, the range was 0~2 m, the accuracy was ±0.5% F.S., and the sampling frequency was up to 100 Hz. Sensors are calibrated before delivery, however, prior to the test, the accuracy of the measurement data will be verified in air and still water. In the research of wave characteristics, Guo [33] selected the sampling frequency of wave elevation gauge as 50 Hz, Liu [34] chose 25 Hz in his study of wave characteristics. Zhang’s [35] results showed that the wave frequency of the free surface with strong turbulence and surface tension is low frequency, around 20 Hz. In this test, the sampling frequency of digital wave elevation gauge is selected to be 10 Hz, 20 Hz, 50 Hz and 100 Hz for comparison. It can be seen from the Figure 2 that the data collected is more complete when the frequency is 100 Hz, so the sampling frequency of this test is determined to be 100 Hz. The sampling time is 3 min, 4 min, 5 min and 6 min for comparison, Table 3 gives the results. It can be seen from the data that when the sampling time is 5 min, the measured data tends to be stable, but in order to ensure higher accuracy of the data, the sampling time is set to 6 min.

### 2.2. Data Processing

According to wave theory, the processing of wave signals in this research is as follows:

(1) Data preprocessing

Signal denoising.

To eliminate the influence of noise, this study uses a five-point cubic smoothing method to denoise the measured wave data. This method can significantly reduce the high-frequency random noise mixed into the test data when it is applied to the time domain data. Reference [36] describes its processing in detail.

Elimination of the trend term.

The method used in this study to eliminate the trend term is the polynomial least squares method, and the calculation principle can be found in reference [37].

(2) Calculation of the wave elements

The characteristic wave height ($H_{1/100}$, $H_{1/10}$, $H_{1/3}$ and $H_{100/100}$), period, and basic elements of the waves are shown in Figure 3 (The serial number in the figure indicates the wave period, and there are 7 wave periods in the figure.). The wave height calculation employs the zero-crossing method, and the calculation of the characteristic wave height is based on the representative wave method [15]. For example, *H*_i_ is the sequence of wave height from the biggest to the smallest, *T*_i_ is the period corresponding to wave height *H*_i_, $H_{1/100}$ is the average of the first 1/100 wave height (*H*_i_), T is the average period,$\;H_{1/100}=\frac{10}{n}\sum_{i=1}^{10/n}H_{i},T=\frac{1}{N}\sum T_{i}$. Similarly, $H_{1/10}$ is the average of the first 1/10 wave height (*H*_i_), $H_{1/3}$ is the average of the first 1/3 wave height (*H*_i_), and $H_{100/100}$ is the average wave height.

(3) In the wave spectrum analysis of the stilling basin with a negative step, this study uses the correlation function method [12] to analyze the power spectrum of the wave signal of the stilling basin with a negative step to obtain the frequency distribution of the waves.

## 3. Experimental Results

### 3.1. Basic Flow Patterns

To make the results more representative, a dimensionless number, the flow-energy ratio *k* ($k=q/\sqrt{g{\left(\Delta h\right)}^{3}}$, where *q* is the unit discharge; Δh is the water level difference between upstream and downstream,$\;\Delta h=h_{0}-h_{2}$; and g is the acceleration of gravity) that can represent hydraulic conditions is introduced here. *d*/*P* is the relative step height (*d* is the height of the step, and *P* is the height of the dam). In this hydraulic model test, it was observed that there are two flow patterns in the stilling basin with a negative step, namely, B-jumps and Wave-jump. The latter appears only when *Q* = 30 L/s and *d*/*P* = 0.0651. The two flow patterns are described below.

#### 3.1.1. B-Jump

When a B-jump occurs, there is an anti-arc section in the spillway at the head of the hydraulic jump position, and the main flow quickly descends after passing the step, forming a local oblique submerged jet in the stilling basin. The interaction between the submerged jet and the water in the stilling pool is very strong. The main flow spreads to the surroundings, and strong shear and turbulence are formed around the main flow. The interaction between the submerged jet and the water body in the stilling pool is very strong. The main flow spreads around, and strong shear and turbulence are formed around the jet, which form a bottom roller after the step, and a large surface roller is formed in the anti-arc section of the spillway and the stilling basin. In general, B-jump can exhibit a mixed flow pattern of oblique submerged jets and submerged hydraulic jumps. When a B-jump occurs, the water surface in the stilling basin is relatively stable, and the energy dissipation rate is high. Therefore, the B-jump is the ideal flow pattern in the stilling basin with a negative step and is the flow pattern needed in the project. The wave characteristics of the B-jump flow pattern in the stilling basin with a negative step are the focus of this research. Figure 4 shows the test photos and schematic diagrams of B-jump under some conditions.

#### 3.1.2. Wave-Jump

When Wave-jump occurs, the main flow passes through the anti-arc section and shoots out from the top of the step. At this time, the water flow bends upwards, forming a wave, similar to a jet stream. After the main flow falls into the water, it forms a surface roller and causes violent water surface fluctuations. There is a bottom roller between the lower part of the main flow and the bottom plate. Since the incident main flow of Wave-jump first floats up and then dives into the water, higher waves occur in the stilling basin to some extent, so the water surface in the stilling basin is not stable, which has a very negative impact on downstream riverbed protection, navigation and other engineering tasks [5]. This kind of flow pattern has also been observed in a stilling basin with a negative step and a flat floor. Scholars such as Kawagoshi [2] and Ohtsu [38] have carried out more research on this flow pattern. Figure 5 shows the test photos of Wave-jump.

### 3.2. Wave Height

Wave height is an important indicator of wave elements and needs to be considered in engineering design. Figure 6 shows the variation in relative wave height *H*_1/100_/*h*_2_ (*h*_2_ is the water depth after the jump) along the stilling basin with a negative step. The abscissa in the figure is the relative position *x*/*L*, *x* is the distance between the wave elevation gauge and the starting point of the stilling basin, and *L* is the length of the stilling basin. The figure shows that under different discharges, the *H*_1/100_/*h*_2_ in the stilling basin shows a gradual attenuation trend as the propagation distance increases. Except W-jump, the *H*_1/100_/*h*_2_ in the standard stilling basin (TP0) is smaller than that in the stilling basin with a negative step (TP1~TP3). With increasing discharge, the degree of fluctuation along the stilling basin gradually increases, and the attenuation speed of the wave height gradually increases along the stilling basin. At the same section position, as the discharge increases, *H*_1/100_/*h*_2_ increases to varying degrees. When $8.0\leq Fr_{1}\leq 13.7$ and 0.0325≤d/P≤0.0651, the concentration of *H*_1/100_/*h*_2_ ranges from 0.05 to 0.45.

In addition, under the same discharge conditions, with increasing relative step height *d*/*P*, the *H*_1/100_/*h*_2_ of the same section position shows a decreasing trend. However, TP3 (*d*/*P* = 0.0650, *Q* = 30 L/s) in the figure shows an inconsistent trend. Since the flow pattern in the stilling pool is a Wave-jump, the water flow does not dive to the bottom after entering the stilling basin but forms a wave upward, which causes the overall wave height in the stilling basin to be too large, and the main impact range is *x*/*L* < 0.4706. By comparison, it is found that *Q* = 50 L/s; when the relative step height increases from *d*/*P* = 0 to *d*/*P* = 0.0651, the maximum reduction in *H*_1/100_/*h*_2_ is 39.08%, which shows that the height of the step has the most obvious impact on the wave height of *H*_1/100_ at this time.

### 3.3. Average Period

The average period *T* (the total time divided by the number of wave height) of the wave is the ratio of the total time to the number of waves, which represents the average speed of the fluctuations. Figure 7 shows the variation in the average period of waves in different stilling basins with a negative step under different discharge conditions. From the figure, we can obtain the following conclusions:

(1) There is no obvious relationship between the average wave period and the step height. Dong and Lai [12] also mentioned in their wave research that water surface fluctuation is relatively complicated and that the conventional average period calculation is not enough to reveal an accurate trend; in-depth research and analysis are needed;

(2) In the B-jump flow pattern, the trend of the average period variation along the stilling basin with a negative step is consistent; generally, it increases first and then decreases along the flow direction when the *Q* ≥ 50 L/s;

(3) As the discharge increases, the self-similarity (coincidence of the curve) of the average period along the flow direction increases because as the discharge increases, the impact of the step height on the waves decreases;

(4) In this study ($8.0\leq Fr_{1}\leq 13.7$ and 0.0325≤d/P≤0.0651), the average period of the waves in the stilling basin with a negative step is concentrated in the range of 0.33~0.71 s.

### 3.4. Wave Probability Density

The probability density of waves can be calculated by the following formula:(1)$$f\left(H_{i}\right)=\frac{1}{\sqrt{2\pi }\sigma }e^{-{\left(H_{i}-\mu \right)}^{2}/2\sigma^{2}}$$
where $f\left(H_{i}\right)$ is the probability density function; μ is the expectation; σ is the standard deviation; and $\sigma^{2}$ is the variance. In the normal distribution, μ describes the position of the central tendency, and σ represents the shape parameter. Here, the probability density of the stilling basin with a negative step is analyzed with discharge *Q* = 70 L/s as an example. Figure 8 shows the probability density distribution of the measured values of waves for four test plans (TP0~TP3) when the discharge *Q* = 70 L/s. The figure shows the following:

(1) The probability density distribution along the flow direction of different stilling basins with a negative step shows strong similarity due to the same flow pattern in the stilling basin with a negative step (B-jump), the same causes of wave formation and the same form of propagation occurring.

(2) As the relative position *x*/*L* increases, the probability density distribution shape gradually develops from “short-fat” to ”thin-tall”, and tends to *y*-axis symmetry, that is, $\sigma^{2}$ decreases and the mean value gradually tends to zero along the flow direction, because the water is viscous, the wave energy is dissipated during the propagation process, the reduction in wave energy causes the degree of fluctuation to decrease, and the degree of wave dispersion decreases along the flow direction. The amplitude of the wave gradually moves closer to the average along the flow direction, and the up and down fluctuations tend to stabilize.

(3) The data from Figure 7 show that in the same discharge conditions, with the increase of relative height of step *d*/*P*, the $\sigma^{2}$ of the same position show a decrease trend, namely the shape of the probability density distribution is more and more tall. This occurs because the distance from the bottom to main flow on the rise with the increase of step height, making the main flow friction and mix with the surrounding water more, the energy dissipation is higher, so the wave energy generated by water at the surface is less, and it fluctuates less.

To further study the extent of wave surface fluctuations, the skewness coefficient *Sk* and kurtosis coefficient *Ku* are introduced to test the normality of the wave data. The calculation formulas of the skewness coefficient *Sk* and kurtosis coefficient *Ku* are:
(2)$$\mathrm{skewness coefficient}\;Sk=\frac{\frac{1}{n}\sum_{i=1}^{n}{\left(H_{i}-\bar{H}\right)}^{3}}{{\left[\frac{1}{n}\sum_{i=1}^{n}{\left(H_{i}-\bar{H}\right)}^{3}\right]}^{\frac{3}{2}}}$$
(3)$$\mathrm{kurtosis coefficient}\;Ku=\frac{\frac{1}{n}\sum_{i=1}^{n}{\left(H_{i}-\bar{H}\right)}^{4}}{{\left[\frac{1}{n}\sum_{i=1}^{n}{\left(H_{i}-\bar{H}\right)}^{2}\right]}^{2}}$$
where $H_{i}$ is the wave surface height and $\bar{H}$ is the mean value of the $H_{i}$. The skewness coefficient (*Sk*) is an indicator used to measure the asymmetry of the probability distribution of a random signal, and the kurtosis coefficient (*Ku*) is an indicator used to measure the steepness or smoothness of the data distribution. For the standard normal distribution, *Sk* = 0, and *Ku* = 3.

The calculation results of the skewness coefficient (*Sk*) and kurtosis coefficient (*Ku*) are shown in Figure 9. The figure shows that for both a standard stilling basin and a stilling basin with a negative step, the larger the relative position *x*/*L* is, the closer the probability distribution of the waves is to the normal distribution (*Sk* = 0, *Ku* = 3). This occurs because the wave generation is the strong turbulence formed after the incident main flow enters the stilling basin; the continuous generation of vortices complicates the water surface fluctuations; the farther the relative position is, the more energy the water flow into the stilling basin can dissipate; and irregularly moving water surface fluctuations are continuously adjusted along the flow direction, making the water flow close to a nonuniform gradual flow, so the dispersion degree of the wave relative to the average value is smaller. A study by Dong [12] et al. found that the wave probability density is more skewed approaching the nose sill and is close to the normal distribution far from the nose sill. The main reason for this is that there is a roller area near the nose sill, so the data have a large degree of dispersion. Sun [39] also mentioned in his paper that the probability density distribution of waves is closer to a normal distribution as the distance from the step increases; when the discharge increases, *Sk* and *Ku* both tend to increase because when the discharge increases, the turbulence of the water entering the basin increases, resulting in more violent water surface fluctuations and poor stability.

The figure also shows that there are no obvious differences between the standard stilling basin (TP0) and stilling basin with a negative step (TP1~TP3). When $8.0\leq Fr_{1}\leq 13.7$ and 0.0325≤d/P≤0.0651, the skewness coefficient (*Sk*) of the wave probability distribution is in the range of 0.0062~0.7535, and the kurtosis coefficient (*Ku*) is 2.8531~4.7516.

### 3.5. Power Spectrum

The frequency-domain properties are usually represented by spectra, which can describe the internal structure of waves. Figure 10 shows the time-domain process and power spectrum of the wave signal at *Q* = 30 L/s and relative position *x*/*L* = 0.047 in the stilling basin with a negative step (TP3). Figure 10a shows that the wave signal in the stilling basin with a negative step is a typical random process signal. The result of Figure 10b shows that the wave signal in the stilling pool is in the low-frequency narrowband noise spectrum [40]. The frequency band is narrow, and the wave energy is concentrated in the low-frequency area (*f* < 10 Hz).

The frequency with the highest spectral density is called the dominant frequency (also known as the dominant frequency or peak frequency) and is denoted as $f_{0}$. In water conservancy projects, it is generally believed that when the main frequency is equal to or close to the natural frequency of the hydraulic structure in the water, resonance is likely to occur, leading to the destruction of the hydraulic structure. Therefore, it is necessary to obtain the distribution relationship of $f_{0}$ through the spectrum analysis method. The distribution of the $f_{0}\;$of waves in the stilling basin with a negative step is shown in Figure 11. From the figure, we can obtain the following:

(1) The $f_{0}$ along the stilling basin with a negative step shows a trend of first increasing and then decreasing, and the maximum value of $f_{0}$ appears in the range of 0.20 < *x*/*L* < 0.45.

(2) The correlation between the $f_{0}$ of the wave and the step height is weak, and the $f_{0}$ values along the flow direction of the standard stilling basin (TP0) and the stilling basin with a negative step (TP1~TP3) are not very different.

(3) When $8.0\leq Fr_{1}\leq 13.7$ and 0.0325≤d/P≤0.0651, the wave energy is mainly concentrated in the low-frequency band, and $f_{0}$ is less than 2.5 Hz.

## 4. Discussion

### 4.1. Calculation of the Characteristic Wave Height

For an engineering designer, it is often necessary to use the values of various characteristic wave heights as design standard according to the project scale. Under the conditions of a certain wave height distribution, there is a certain connection between various characteristic wave heights. After statistical calculation of the wave height data of different measuring points in the stilling basin with a negative step, the relationships between the different characteristic wave heights $H_{1/100}$, $H_{1/10}$, and $H_{1/3}$ and $H_{100/100}$ are:(4)$$H_{1/100}=\left(2.299\sim 3.302\right)H_{100/100}$$
(5)$$H_{1/10}=\left(2.160\sim 3.002\right)H_{100/100}$$
(6)$$H_{1/3}=\left(2.093\sim 2.812\right)H_{100/100}$$

However, there is no theoretical or empirical formula for calculating the characteristic wave height in a stilling basin with a negative step. When the height of the sidewall of the stilling basin is designed, if the wave height is not considered, it is likely to cause problems such as water flow over the sidewall and scouring of the bank slope; if the height of the sidewall is considered to be too high, it would increase the project cost. Therefore, accurate estimation of the wave height in the stilling basin with a negative step is particularly important in actual engineering.

In a study of posthydraulic-jump waves, Abou-Seida [41] selected the wave height $H_{i}$, wave period T, upstream water depth $h_{0}$, downstream water depth $h_{2}$, canal bottom slope θ, and the acceleration of gravity g as variables; after dimensional analysis, the relationship between the wave height and variables was obtained:$\frac{H_{i}}{h_{2}}=f\left(\frac{H}{gT^{2}},\frac{v_{1}^{2}}{2gh_{0}},\theta \right);\;$finally, after the sink test, it was concluded that the effective wave height increases with increasing dynamic factor $\lambda =v_{1}^{2}/2gh_{1}$. Wu et al. [11]. proposed a relationship between the relative maximum wave height and Froude number after analyzing measured data in their study of wave characteristics after a hydraulic jump:$\;H_{max}/h_{1}=0.369\left(Fr_{1}-1\right)$.

Based on the above analysis, it can be considered that the characteristic wave height and Froude number of the stilling basin with a negative step before the jump $Fr_{1}$, the water depth before the jump $h_{1}$, the height of the step d, the water depth after the jump $h_{2}$, the acceleration of gravity g and the water density ρ are related. Here, taking the calculation of the $H_{1/100}$ wave height as an example, the $H_{1/100}$ wave height can be written as a general function expression:(7)$$H_{1/100}=f\left(Fr_{1},h_{1},h_{2},d,\rho ,g\right)$$

There are 7 physical quantities in the above formula, including 6 independent variables. We choose the three physical quantities $h_{2}$, g and ρ as the basic physical quantities. According to the π theorem, the following can be easily obtained:(8)$$\frac{H_{1/100}}{h_{2}}=f\left(Fr_{1},\frac{h_{1}}{h_{2}},\frac{d}{h_{2}}\right)$$
where$\;Fr_{1}=\frac{v_{1}}{\sqrt{gh_{1}}}$. After regression analysis is performed on the data, the empirical calculation formula for the relative wave height $\frac{H_{1/100}}{h_{2}}$ of the stilling basin with a negative step under B-jump conditions is obtained as:(9)$$\frac{H_{1/100}}{h_{2}}=0.6175Fr_{1}{}^{-0.1711}{\left(\frac{d}{h_{2}}\right)}^{-0.2333}{\left(\frac{h_{1}}{h_{2}}\right)}^{-0.311}\;st.\;\{\begin{array}{c} 8.0\leq Fr_{1}\leq 13.7\\ 0.0325\leq d/P\leq 0.0651 \end{array}$$

Figure 12 is the fitting curve of the test data. According to the calculation results of the statistical error analysis index, the correlation coefficient between the predicted value of the empirical calculation formula and the test value is $R^{2}=0.905$, the average absolute percentage error is MAPE=3.85%, and the accuracy is high.

### 4.2. Effect of the Flow Energy Ratio on the Wave Characteristics at the End of the Stilling Basin

In actual engineering, it is always desirable for the outflow of the stilling basin with a negative step and the downstream water flow to smoothly connect to avoid scouring damage to the bank slope. Therefore, it is of great engineering significance to study the wave characteristics at the end of the stilling basin with a negative step. The wave standard deviation $\sigma_{w}$ reflects the distribution characteristics of the dispersion degree of the wave relative to the average wave height, that is, the position where the greater the standard deviation is, the greater the fluctuation of the wave is. The wave standard deviation can be calculated according to the following formula:(10)$$\sigma_{w}=\sqrt{\frac{1}{n}\overset{n}{\underset{i=1}{\sum }}{\left(H_{i}-\bar{H}\right)}^{2}}$$
where $H_{i}$ is the instantaneous value of the water surface, $\bar{H}$ is the average value of the wave surface, and *n* is the number of samples. The wave data are uniformly taken from the data at the location of the wave elevation gauge before the end sill. To make the results more representative, the standard deviation is processed as a dimensionless number $\sigma_{w}/h_{2}\;$($h_{2}$ is the water depth after the jump) used to represent the fluctuation characteristics of the characteristic wave at the end of stilling basin with a negative step.

Since the B-jump flow pattern is the most important and most frequently occurring flow pattern in the stilling basin with a negative step, in the stilling basin with a negative step, the fluctuation characteristics of waves at the end of the stilling basin with a negative step ($\sigma_{w}/h_{2}$) are mainly discussed. Due to the essential difference between the flow pattern in the standard stilling basin (TP0) and that of the stilling basin with a negative step (TP1~TP3), it is necessary to separately discuss them. Figure 13a shows the fitting curve of the energy ratio *k* and $\sigma_{w}/h_{2}$ under a B-jump, and the empirical calculation formula is:(11)$$\frac{\sigma_{w}}{h_{2}}=0.708k+0.0011\;st.\;\{\begin{array}{c} 8.0\leq Fr_{1}\leq 13.7\\ 0.0325\leq d/P\leq 0.0651 \end{array}$$

According to the calculation result of the statistical error analysis index, the correlation coefficient between the predicted value of the empirical calculation formula and the experimental value is $R^{2}=0.8733$, the average absolute percentage error is MAPE=12.62%, and the accuracy is high.

Figure 13b shows the relationship between $\sigma_{w}/h_{2}$ and *k* in the standard stilling basin. The figure clearly shows that there is the following linear relationship between $\sigma_{w}/h_{2}$ and *k*:(12)$$\frac{\sigma_{w}}{h_{2}}=0.5246k+0.0119$$

In summary, for both a stilling basin with a negative step and a standard stilling basin, there is a good linear relationship between $\sigma_{w}/h_{2}$ and *k*, and when their k values are close, the $\sigma_{w}/h_{2}$ of the stilling basin with a negative step is smaller than that of the standard stilling basin. This is due to the existence of the step in the stilling basin with a negative step, which increases the diffusion distance of the main flow into the basin, intensifies the shearing effect with the surrounding water body, and exhibits more sufficient energy dissipation from the water flow. Therefore, the fluctuation degree in the stilling basin with a negative step is smaller than that in the standard stilling basin.

### 4.3. Effect of the Relative Step Height on the Wave Characteristics at the End of the Stilling Basin

Figure 14 shows the relationship between the relative step height *d*/*P* and $\sigma_{w}/h_{2}$. The figure shows that with increasing *d*/*P*, $\sigma_{w}/h_{2}$ shows an approximately linear decreasing trend, and with increasing discharge, $\sigma_{w}/h_{2}$ increases under the same step height. When *d*/*P* = 0.065 and *Q* = 30 L/s, there is an abnormal point in the trend, mainly because the flow pattern in the stilling basin with a negative step is Wave-jump. In this flow pattern, the main flow enters the stilling basin and forms an upward jet, which causes the overall wave value in the stilling basin to be higher and the stability to be poor.

Further analysis shows that under different discharge conditions, there is a good linear relationship between $\sigma_{w}/h_{2}$ and *d*/*P* under different flow conditions. According to the linear relationship between $\sigma_{w}/h_{2}$ and *d*/*P*, as the discharge increases, the slope and intercept increase.

## 5. Conclusions

In this study, the waves in a stilling basin with a negative step were studied through hydraulic model tests. Based on wave theory, the trends of wave height and other elements along the flow direction were analyzed, and the effects of the flow-energy ratio *k* and relative step height (*d*/*P*) on the wave characteristics were discussed. The following conclusions were drawn:

(1) Two typical flow patterns in the stilling basin with a negative step are observed in the experiment, B-jumps and Wave-jump. The Wave-jump has a greater influence on the wave characteristics than the B-jump. For both Wave-jump and B-jumps, the *H*_1/100_/*h*_2_ increase with discharge and decrease with relative step height *d*/*P.* There is no obvious relationship between the average wave period and the step height, but the change in the average period shows a trend of first increasing and then decreasing along the flow direction when *Q* ≥ 50 L/s.

(2) The probability density distribution of the wave gradually tends to the standard normal distribution as the relative position *x*/*L* increases. The dominant frequencies $f_{0}$ of the wave in the stilling basin with a negative step are all less than 2.5 Hz.

(3) According to the experimental data, the conversion relationships between different wave heights are obtained. Based on the π theorem and regression analysis, the empirical calculation formula for the relative characteristic wave height $H_{1/100}/h_{2}$ is obtained. The dimensionless standard deviation $\sigma_{w}/h_{2}$ at the end of the stilling basin with a negative step is linearly related to the flow-energy ratio k. For the B-jump, $\sigma_{w}/h_{2}$ shows a good linear relationship with the relative step height *d*/*P*.

## Figures and Tables

**Figure 1 entropy-24-00445-f001:**
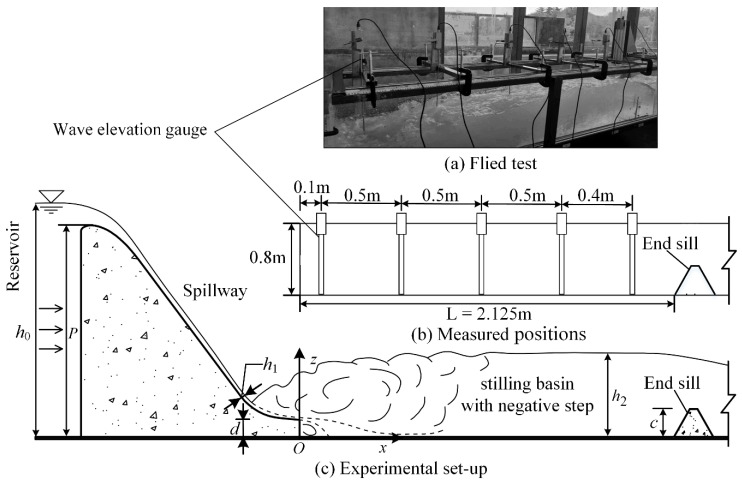
Experimental setup.

**Figure 2 entropy-24-00445-f002:**
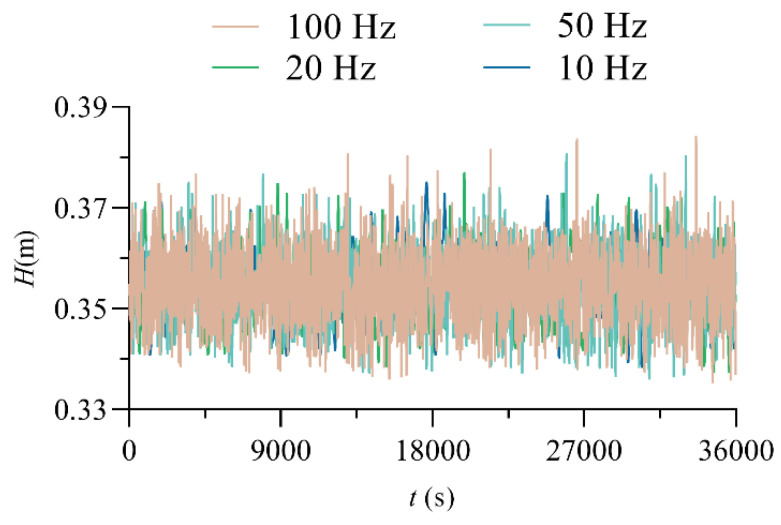
Wave time-domain diagram with different sampling frequencies. (TX0, Q = 30 L/s, *x*/*L* = 0.8941).

**Figure 3 entropy-24-00445-f003:**
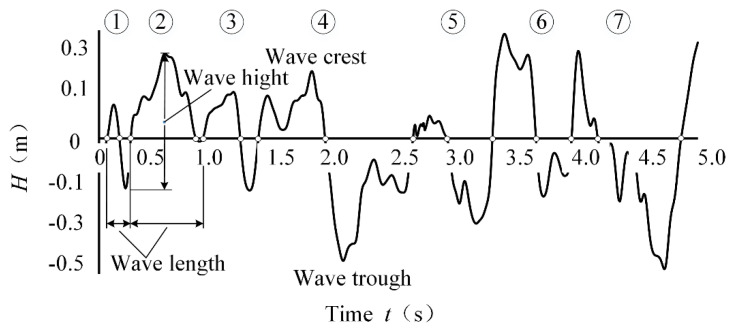
Schematic diagram of basic wave elements.

**Figure 4 entropy-24-00445-f004:**
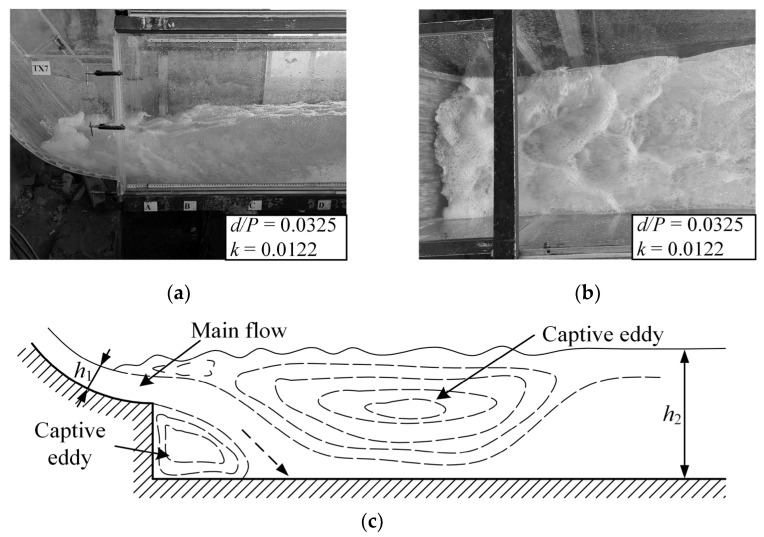
B-jump. (**a**) Front view; (**b**) Top view; (**c**) Schematic view.

**Figure 5 entropy-24-00445-f005:**
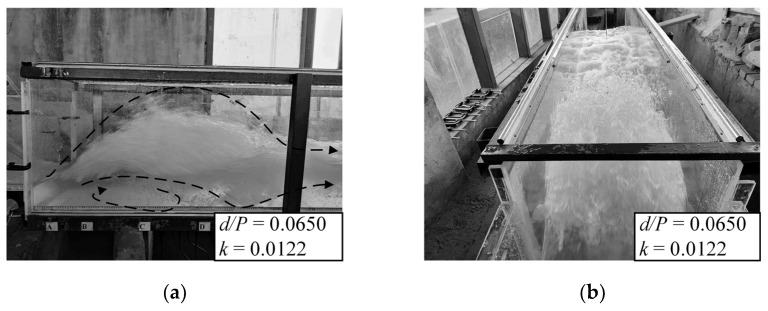
Wave-jump. (**a**) Front view; (**b**) Top view; (**c**) Schematic view.

**Figure 6 entropy-24-00445-f006:**
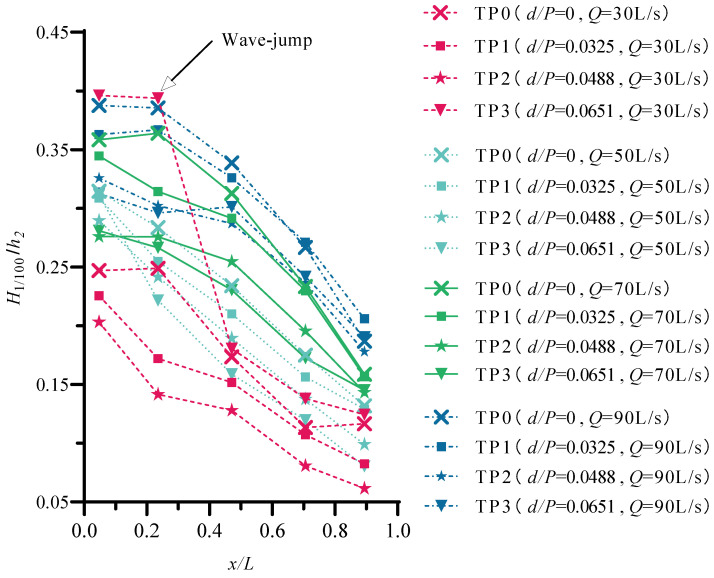
The distribution of the wave height along the stilling basin.

**Figure 7 entropy-24-00445-f007:**
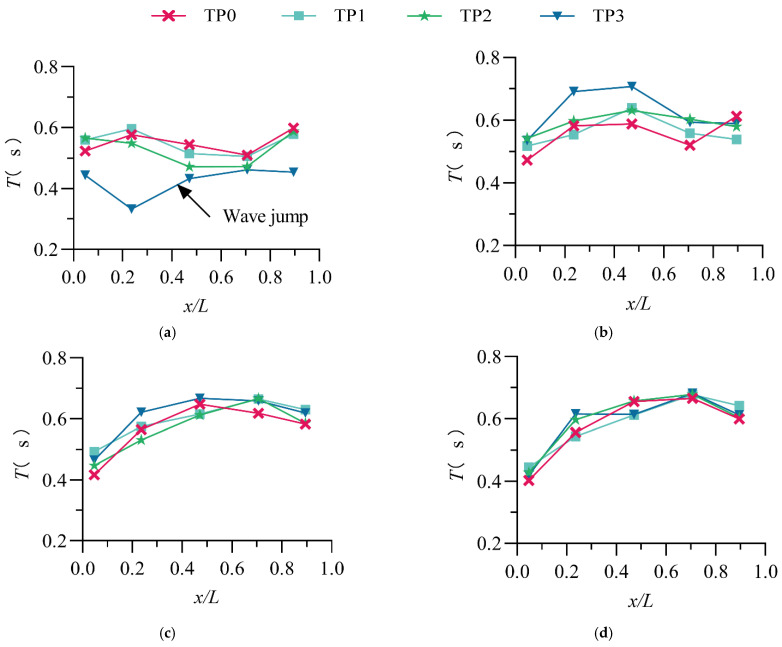
The distribution of the average period along the stilling basin. (**a**) *Q* = 30 L/s; (**b**) *Q* = 50 L/s; (**c**) *Q* = 70 L/s; (**d**) *Q* = 90 L/s.

**Figure 8 entropy-24-00445-f008:**
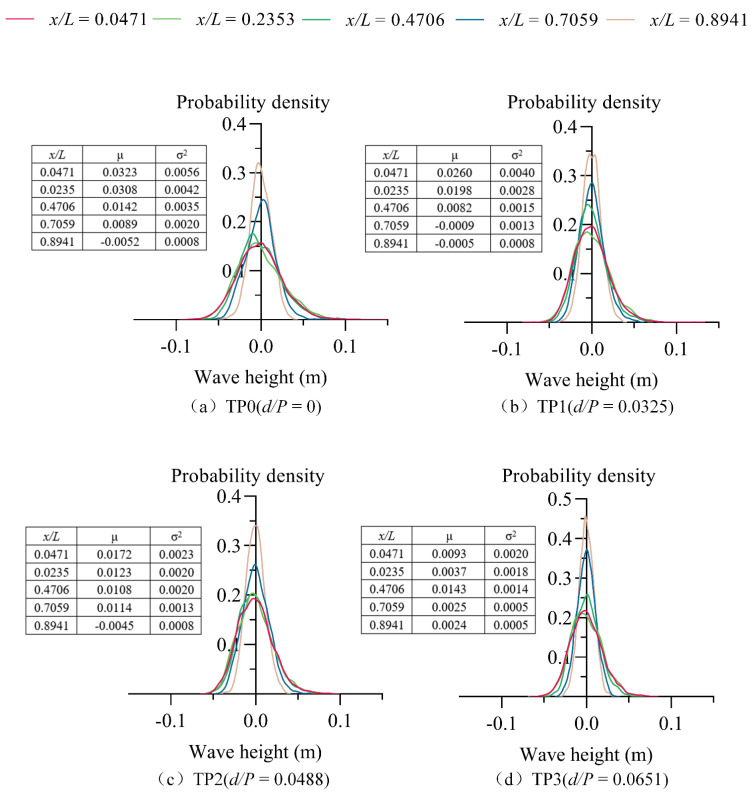
The distribution of the wave probability density (*Q* = 70 L/s).

**Figure 9 entropy-24-00445-f009:**
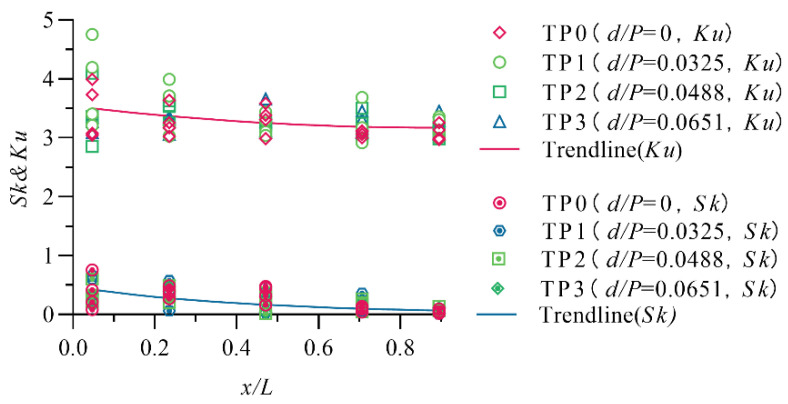
The distribution of Ku and Sk along the stilling basin with a negative step.

**Figure 10 entropy-24-00445-f010:**
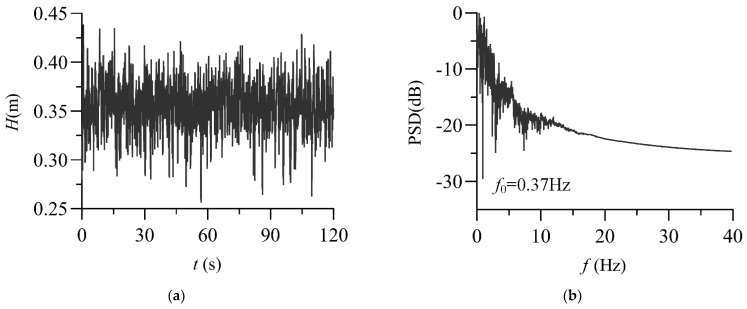
Power spectrum analysis of the wave signal (TP3, *Q* = 30 L/s, *x*/*L* = 0.0471). (**a**) Wave time-domain diagram; (**b**) Wave power spectrum.

**Figure 11 entropy-24-00445-f011:**
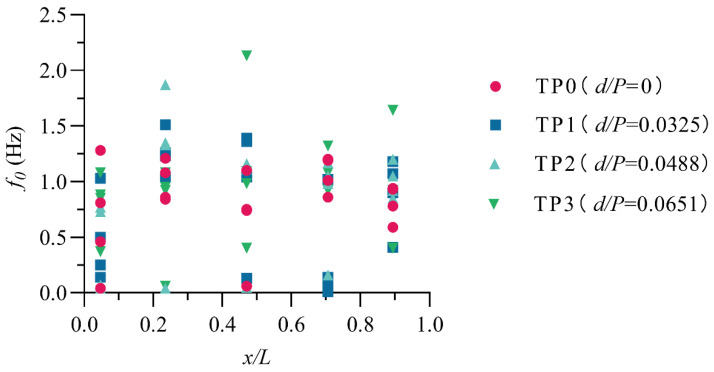
The distribution of the dominant frequency along the stilling basin.

**Figure 12 entropy-24-00445-f012:**
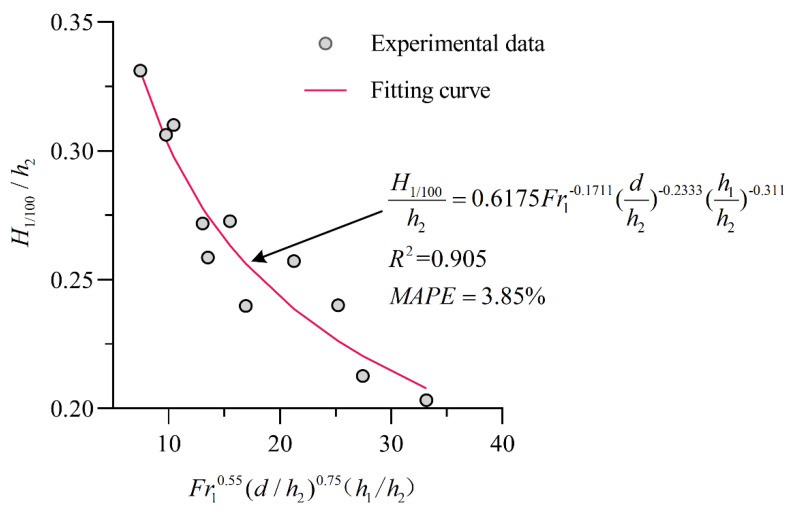
Experimental data and fitting curve.

**Figure 13 entropy-24-00445-f013:**
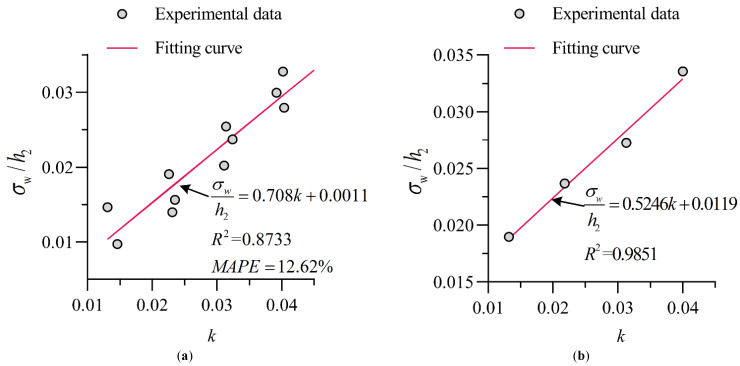
Experimental data and fitting curve. (**a**) Stilling basin with a negative step (TP1~TP3); (**b**) Standard stilling basin (TP0).

**Figure 14 entropy-24-00445-f014:**
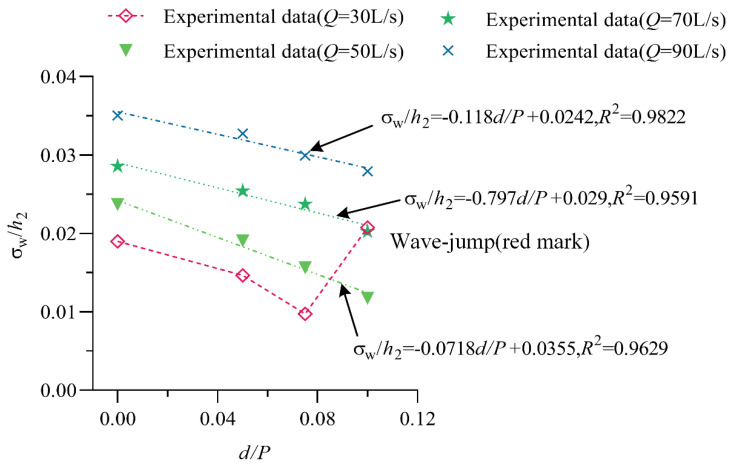
The relationship between $\sigma_{w}/h_{2}$ and d/P.

**Table 1 entropy-24-00445-t001:** The representative large hydropower project in the world using the stilling basin with a negative step.

No.	Name	Dam Height (m)	Design Discharge (m^3^/s)	Country
1	Sayano-Shushenskaya	242.0	13,600	Russia
2	Tehri Dam	260	12,200	India
3	TaSang Dam	227.5	−	Burma
4	Xiangjiaba	162.0	41,200	Chian
5	Huangjinping	95.5	5650	Chian
6	Jin’anqiao	160.0	11,668	Chian
7	Guanyinyuan	159.0	16,900	Chian
8	Liyuan	155.0	11,361	Chian
9	Guandi	168.0	14,000	Chian
10	Tingzikou	110.0	34,500	Chian

**Table 2 entropy-24-00445-t002:** Parameters of stilling basin.

Test Plan	*D* (m)	Test Plan	*D* (m)
TP 0	0.000	TP 1	0.075
TP 2	0.050	TP 3	0.100

**Table 3 entropy-24-00445-t003:** Comparison of wave elements at different sampling times. (TX0, Q = 30 L/s, *x*/*L* = 0.8941).

Wave Elements	Sampling Times				
	2 min	3 min	4 min	5 min	6 min
$\bar{H\;}$(m)	0.0430	0.4356	0.4311	0.4255	0.4258
*H*_1/100_ (m)	0.0392	0.0425	0.0414	0.0411	0.0414
*T* (s)	0.6653	0.7123	0.6932	0.6993	0.6991

## Data Availability

Not applicable.

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
