# Peer review of "Experimental Study on Wave Characteristics of Stilling Basin with a Negative Step"

_entropy, 2022, doi:10.3390/e24040445_

Round 1
Reviewer 1 Report
Comments and Suggestions for Authors are given in the word file.

Author Response
Response to Reviewer 1 Comments
Point 1: Please check the English writing of the full paper carefully .“because it can avoid atomization and can decrease scouring problems.” ->“because it can avoid atomization and decrease scouring problems.” “Although stilling basins with a negative step have attracted many attention from researchers, few researchers have focus on the wave characteristics.” ->“Although stilling basins with a negative step have attracted much attention from researchers, few researchers have focused on the wave characteristics.” Line 98 “Huang studied the characteristics of roll-wave based on numerical investigation, such wave–wave interactions, generality, and spectrum.” ->“Huang studied the characteristics of roll-wave based on numerical investigation, such as wave–wave interactions, generality, and spectrum.” “the position where the greater the standard deviation is, the greater the fluctuation of the wave”-> “the position where the greater the standard deviation is, the greater the fluctuation of the wave is”. “the farther the relative position is, the more the water flow into the stilling basin can dissipate energy;” ->“the farther the relative position is, the more energy the water flowed into the stilling basin can dissipate;”
Response 1: We quite agree with you ,I read and checked the full text carefully and revised some English writing.
Point 2: Line130: “Error! Reference source not found”. Did you make mistakes there?
Response 2: It is because the hyperlinks could not be displayed when converting to
PDF.
Point 3: Line138: “Table 1. This is a table. Tables should be placed in the main text near to the first time they are cited”. Is this the name of the table?
Response 3:Sorry , the name of the table is “Parameters of stilling basin “,I modified it.
Point 4: The second conclusion in Average Period of Section 2 is not so obvious when Q is equal to 30L/s, should this conclusion be revised?
Response 4: We quite agree with you, I modified this conclusion and added a discharge restriction condition in the conclusion
Point 5: I was confused about ??1, how do you determine its range.
Response 5:Value of h1, which are the water depths at the starting position of the hydraulic jump(Fig.1, Fig.3, Fig.4), were measured in model tests, and velocity u1 was computed by means of the continuity law,, g is the acceleration of gravity.
|
Fig. 1.Experimental set-up |
|
Fig. 3. B-jump(c)Schematic view |
|
Fig. 4. B-jump(c)Schematic view |
Point 6: The fitting formula in Figure 13 may have some mistakes, please confirm if it is correct.
Response 6: I rechecked the data and found there was no problem with it. The following is my test data.
|
d/P |
/ |
|||
|
30 L/s |
50 L/s |
70 L/s |
90 L/s |
|
|
0 |
0.0190 |
0.0237 |
0.0286 |
0.0350 |
|
0.05 |
0.0146 |
0.0191 |
0.0254 |
0.0327 |
|
0.075 |
0.0097 |
0.0156 |
0.0237 |
0.0299 |
|
0.1 |
0.0207 |
0.0117 |
0.0202 |
0.0279 |
Point 7: It is innovative to study the stilling basins with a negative step by wave theory, what is the advantage by adopting this methods compared with other ones?
Response 7: Wave signal belongs to random signal and should be processed according to the processing method of random signal. The characteristic wave height, probability density distribution and spectrum characteristics obtained based on wave theory are more representative.
Finally, thank you very much for your time and valuable suggestions on my manuscript. I have made improvements to my manuscript according to your suggestions. I hope that this improved manuscript can be recognized by you.
Wish you all the best!

Reviewer 2 Report
This paper was dealing with wave characteristics in a stilling basin. The authors measured wave heights, periods and energies. The reviewer did not understand why those experiments are needed.
The wave profiles observed in the basin looks like a three-dimensional shape in Figures 3 and 4. But the authors carried out experiments in a two-dimensional tank.
There is no calibration data for the digital wave elevation gauges.
The uncertainty assessment data should be included.
Author Response
Point 1: This paper was dealing with wave characteristics in a stilling basin. The authors measured wave heights, periods and energies. The reviewer did not understand why those experiments are needed.
Response 1: I'm sorry that I don't quite understand what you mean, but I hope my answer can be close to what you want. Water surface fluctuation is a common phenomenon in a stilling basins.The standard stilling basins were mainly used for low-head flood discharge and energy dissipation in the past, so the water surface fluctuations were small and did not affect the safe operation of the project, there are few studies on water surface. However, a stilling basin with a negative step is mainly used in high-head and large-flow flood discharge projects, the water surface fluctuates drastically during flood discharge, and the water surface problem cannot be ignored. The fluctuation of the water surface in the stillin basin can be regarded as wave, therefore this experiment studies the fluctuation of the water surface from the angle of wave. Wave height and period are the basic elements to describe waves, which can help us to have a deeper understanding of the water surface fluctuation in the stilling basin, at the same time, it can also provide suggestion for practical engineering design.
Point 2: The wave profiles observed in the basin looks like a three-dimensional shape in Figures 3 and 4. But the authors carried out experiments in a two-dimensional tank.
Response 2: I quite agree with you, wave is three-dimensional, there are different directions of propagation, a lot of wave research is to study the direction spectrum of wave, this manuscript is mainly to study the characteristics of wave on the axis of the stilling basin.
Point 3: There is no calibration data for the digital wave elevation gauges.
Response 3: We quite agree with you, I have made a supplement in this manuscript.
Point 4: The uncertainty assessment data should be included.
Response 4: We quite agree with you, I have made a supplement in this manuscript.
Finally, thank you very much for your time and valuable suggestions on my manuscript. I have made improvements to my manuscript according to your suggestions. I hope that this improved manuscript can be recognized by you.
Wish you all the best!

Round 2
Reviewer 2 Report
The manuscript was revised by the reviewer's comments.